# Finite Element Modelling of a Gram-Negative Bacterial Cell and Nanospike Array for Cell Rupture Mechanism Study

**DOI:** 10.3390/molecules28052184

**Published:** 2023-02-26

**Authors:** Majedul Islam, Falah Sahal S. Aldawsari, Prasad K. D. V. Yarlagadda

**Affiliations:** 1School of Mechanical Medical and Process Engineering, Faculty of Engineering, Queensland University of Technology, Brisbane, QLD 4001, Australia; 2Dean (Engineering), University of Southern Queensland, Springfield Central, QLD 4300, Australia

**Keywords:** nanotexture, 3D finite volume model, gram-negative bacteria, cell-rupture mechanism

## Abstract

Inspired by nature, it is envisaged that a nanorough surface exhibits bactericidal properties by rupturing bacterial cells. In order to study the interaction mechanism between the cell membrane of a bacteria and a nanospike at the contact point, a finite element model was developed using the ABAQUS software package. The model, which saw a quarter of a gram-negative bacteria (*Escherichia coli*) cell membrane adhered to a 3 × 6 array of nanospikes, was validated by the published results, which show a reasonably good agreement with the model. The stress and strain development in the cell membrane was modeled and were observed to be spatially linear and temporally nonlinear. From the study, it was observed that the bacterial cell wall was deformed around the location of the nanospike tips as full contact was generated. Around the contact point, the principal stress reached above the critical stress leading to a creep deformation that is expected to cause cell rupture by penetrating the nanospike, and the mechanism is envisaged to be somewhat similar to that of a paper punching machine. The obtained results in this project can provide an insight on how bacterial cells of a specific species are deformed when they adhere to nanospikes, and how it is ruptured using this mechanism.

## 1. Introduction

The formation of bacterial microcolonies from a biochemical cascade, called ‘biofilm’ [1], on a solid surface (such as of an implant inside the human body, or structures for human use), poses a considerable threat of pathogenic infections [2]. An optimal and lasting solution against these notorious species using a chemical approach is challenging because of the growing biochemical resistance over time [3]. Therefore, it is highly desirable that the solid surface itself be either antibacterial and/or resistant to bacterial adhesion [4,5,6]. Being inspired from the natural antibacterial surfaces, such as the insect wings, plant leaves, and animal skin (see Figure 1a) [7,8,9,10], nanotextured surfaces, fabricated using various chemical and mechanical techniques, were found to be notably antibacterial against various bacteria strains (see Figure 1b) [6,11,12,13,14,15]. 

Bacteria is a single prokaryotic cell organism, bounded by a sticky layer of sugar polymers or polysaccharides, called a capsule, around the membrane (see Figure 1c) [16]. The layer underneath the cell wall of the bacterium is the plasma membrane. The plasma membrane is pushed against the cell wall by an internal force, called turgor pressure [17]. Bacteria can be cylindrical, comma-form, corkscrew, spiral, or spherical. According to the composition and structure of the cell wall, bacteria can be also of -positive and Gram-negative. Gram-negative bacteria, such as *Escherichia coli* (*E. coli*), have an outer membrane. Conversely, Gram-positive bacteria, such as *Streptococcus pneumonia*, are surrounded by several layers of peptidoglycan, which are thicker than the cell membranes of Gram-negative bacteria [16]. The interaction between the bacteria and the nanotextured surface causes deformation to this membrane leading to a fatal rupture, as defined in published literature [18], is known as antibacterial or bactericidal properties. 

Nanotexturing is a surface modification technique, performed by various bottom-up or top-down techniques [19], for producing an array of nanoscale protrusions, usually with height, radius, and spacing dimensions of 100 nm or less [20,21] on a material surface. The range of materials includes, but is not limited to, metals and metal alloys [15,22,23,24,25,26,27,28,29], metalloids [10,30,31,32,33], and polymers [34,35,36,37,38]. The bactericidal efficiency of a nanotextured surface is found to be sensitive to the texture’s dimensions (height, diameter, and spacing) [39,40] and characteristics, such as wettability [28,29,32,33,41,42], and the variety of bacterial strains (Gram-positive or negative) [36,43] as confirmed experimentally [10,32,33,36,37,38,43,44,45,46,47,48,49,50,51,52] and computationally [18,19,53,54,55,56,57,58]. An ample number of experimental studies could be found in the open literature focusing mostly on the design of a nanotextured surface for antibacterial application. The experimental study of the rupture mechanism of bacterial cells through their interaction with nanospikes might be possible using high-tech equipment with real time monitoring capabilities, however, to the authors’ best knowledge, no such studies could not be found in the open literature. The mechanism study is possible computationally by using a suitable simulation software package installed on a personal computer or laptop. However, the computational studies on interaction mechanisms are very few, as discussed below, and almost all of these studies are explorations of nature that investigated the bactericidal efficiency of a certain type of nanotexture with a view that “one size fits for all”.

**Figure 1 molecules-28-02184-f001:**
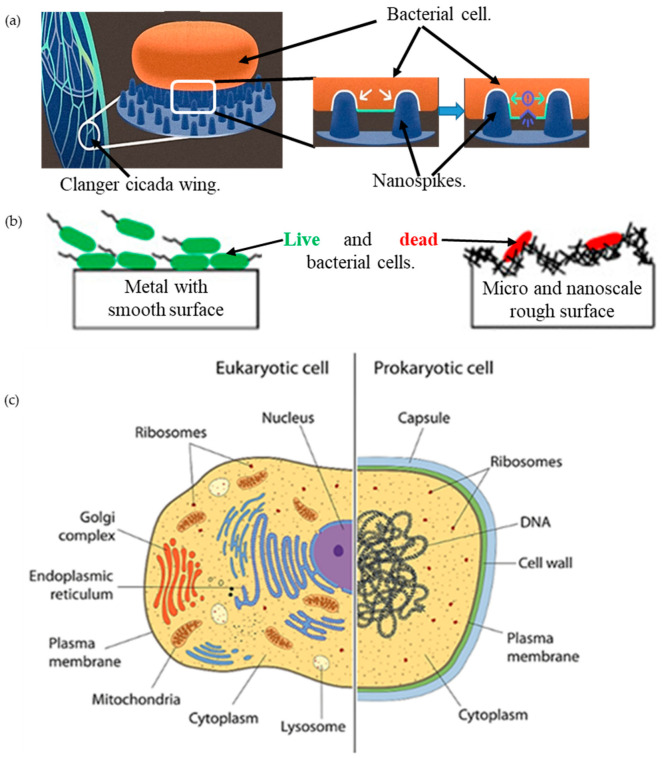
(**a**) Bactericidal ability of the clanger cicada wing, adapted from [59]; (**b**) bacterial interaction on a smooth surface and a micro-/nanoscale rough surface, adapted from [24]; (**c**) bacterial cell versus human cell [60].

Watson, et al. [53] developed a mathematical model based on surface energies, and showed that the nanospike radius should be higher than the minimum required radius, nanospike spacing must be much less than the dimension of the bacterial cell in order to allow the cell to bound to several nanospikes, and the length of the nanospikes must be higher than the required distance to pull the wall of a bacterial cell along the nanospike shaft. Li [54] and Li and Chen [55] found from their thermodynamic models that it is the energy balance between adhesion and the deformation of the cell membrane that makes the bacterial cell adhere to the nanospike surfaces. This adhesion is suspended for nanospikes with radii less than the critical value, while nanospikes beyond the critical radius but with small spacing increases the bacterial adhesion. By analyzing the stretching factors for the wall of the Gram-positive and Gram-negative bacterial cells, Xue, et al. [56] observed a relationship between bacterial cell wall stretching and the nanopattern of clanger cicada wings. Pogodin, et al. [18] observed from their biophysical model that a cell’s rigidity and mechanical properties are important factors for the determination of the resistance and sensitivity of the bacterial cells when they are in contact with cicada wings. Velic, et al. [57] observed that the local stresses due to the bacteria deformation are enough to cause the rupture and death of the bacteria; and the most effective ways for improving the bactericidal efficiency are decreasing the radii of pillars and enhancing the attractive interaction. Velic, et al. [58] further observed that nanopatterned structures are unable to predominantly kill the bacteria by rupture, however, as had been previously believed. In fact, the deformation of a bacteria’s surface around the heads of pillars is more critical for this purpose, as this location is that of maximum deformation. This location was found to have a substantial amount of in-plane strain due to the deformation of the cell, and hence the rupture and penetration point of the cell. The obtained results also indicated that the deformation of multilayered bacteria increases by interaction with nanopattern surfaces, which have smaller pillar radii and spacing. In another study, Velic, et al. [19] observed that the bactericidal performance can be increased by decreasing the nanospikes’ tip radii, and by increasing the ratio between the radius and spacing. 

As has been mentioned earlier, all of these studies investigated the bactericidal properties of their developed nanotextured surfaces using a random bacterial strain, which, therefore, might not be equally effective for a different strain. Very few studies could be found in the open literature that studied the interaction mechanism of a certain bacterial strain with nanospikes in order to design strain-specific bactericidal nanotextured surface. Cui, et al. [61] developed a 3D finite element model of an *E. coli* bacterial cell, and studied its mechano-bactericidal mechanism with a nanostructured surface. The mechanical properties of an *E. coli* cell membrane was investigated using an atomic force microscopy (AFM), and the result was used to validate their simulation model. They found that the bacterial cell receives a higher stress and deformation when located on the nanospike’s surface and, therefore, a two-stage model was developed to take the effects of elastic and creep deformation into consideration. The simulation results indicated that for a bacterial cell that is adhered to the nanospikes, the location of maximum stress and strain is at the contact line of the liquid and the bacterial cell and nanospike surfaces. In order to investigate the rupture mechanism of different bacterial strains, a 3D finite volume model of a Gram-negative bacterial strain was developed as detailed in the current study. For validation purposes, the *E. coli* strain used inCui, et al. [61] was used in the simulation model. This peer reviewed model will be central to vigorously study the bacterial cell rupture mechanism, and to aid in the design of a strain specific bactericidal metal implant, and adapt the model further for similar bacterial strains.

## 2. Results and Discussions

The extracted results are presented in the form of contours and plots, as illustrated in Figure 2 and Figure 3, respectively.

Figure 2 shows that the cell membrane, nanospike, and surrounding liquid form a three-phase contact point, and a contact line can be visualised along the nanospikes array parallel to the *y*-axis. The highest von Mises stress (see Figure 2a) on the cell membrane around the contact points along this contact line was estimated to be around 6.97 MPa, which lies between the cell wall critical stress (5 MPa) and the tensile strength (13 MPa), thus leading to the creep deformation of the cell wall. A higher strain (see Figure 2b) could be noticed at the local regions around this contact line and the nearby suspended regions of the cell than at the contact point itself, leading to a transfer of different stresses from these local regions to the contact line. The maximum value of displacement for the cell wall is around 0.025 μm, which occurs on the upper regions of the cell, as these regions are free to translate along the *x*-axis. The elastic strain energy density per unit of volume, as calculated using Equation (1) [62], of the cell wall was also observed to be the highest (0.985 MPa), as can be seen in Figure 2c. If cell rupture is initiated, it is expected to occur along this three-phase contact line. Somewhat similar to that of the punch indentation mechanism, a nanospike at the contact point may penetrate or puncture the cell wall and pass through it with the adhered part of the cell membrane on its tip.
(1)$$E_{S}=\frac{1}{4}\left[(\sigma_{r}\;:\;\varepsilon_{r}+\sigma_{i}\;:\;\varepsilon_{i}\right)+\sqrt{{\left(\sigma_{r}\;:\;\varepsilon_{r}-\sigma_{i}\;:\;\varepsilon_{i}\right)}^{2}+{\left(\sigma_{r}\;:\;\varepsilon_{i}+\sigma_{i}\;:\;\varepsilon_{r}\right)}^{2}}];$$
where, *σ_r_*, *σ_i_* are the real and imaginary components of the eigenstress tensor, respectively, and *ε_r_*, *ε_i_* are the real and imaginary components of the eigenstrain tensor, respectively.

The trend in stress, strain, and temporal dissipated energy are presented in Figure 3a–d. In Figure 3a, the stress-strain curve shows the cell membrane behaves like an elastic material, and within the biological tissue zone as expected. This approves the defined elastic behavior for the cell wall. At the maximum stress (about 7 MPA), the strain is about 0.275. The stress and strain variation over time in the cell membrane were assumed to be nonlinear, and modelled accordingly, as can be seen in Figure 3b. The stress and strain development in the cell membrane around the contact point in the first second is almost linear with time, and the trend becomes faster around 0.5 s. The strain energy and the friction dissipation energy (refer to Figure 3c,d respectively) rate over the first second of the contact were found to increase non-linearly. The heat flux density generated by the interface element due to frictional heat generation is given in ABAQUS by Equation (2) [62]. The maximum strain energy over the simulation time (1 s) was estimated to be 0.8 MJ. The friction between the spike and the cell membrane increases with the deformation over time, which is reasonable as the contact area increases gradually during the expected indentation. The friction was observed to be increased rapidly just after 0.6 s of contact, which is comparable with the faster rate of stress and strain. The maximum dissipated energy was estimated to be about 0.15 MJ. This first 0.5 to 0.6 s of contact with the nanospike are crucial to develop sufficient stress and strain to initiate rupture within the cell membrane, and subsequent lysis of the bacterial cell afterward. However, further study is required to explore how the cell membrane of an *E. coli* interacts with the nanospikes of varying tip geometry, density (spacing), length, and flexural strength, and the orientation of the cell itself. In addition to these, being a biological element, how quickly a certain bacterial strain can repair or regenerate its damaged membrane before lysis is also important to consider.
(2)$$q_{g}=\eta \tau \dot{s}=\eta \tau \frac{\Delta s}{\Delta t};$$
where, *τ* is frictional stress, Δ*s* is the incremental slip, and Δ*t* is the incremental time.

## 3. Computational Modeling

The finite element software, ABAQUS/CAE2017 (Dassault Syste`mes Simulia Corp., Johnston, RI, USA), was used for this work. The standard core of this software was used for producing the bacterial model and defining the simulation set up, as explained in detail in the following sections. 

### 3.1. Modeling the Cell and the Nanospike Geometry

The *E. coli* cell modelled for this study is shown in Figure 4. The details of the modeled cell membrane and the nanospikes are given in Table 1.

The *E. coli* cell has a capsule geometry consisting of a cylindrical main body and two hemispherical ends (see Figure 4a,b). The top left quarter of the cell was selected for producing a 3D geometry (see Figure 4c) that sits on 3D nanospikes. The cell wall was generated using the 3D deformable shell with revolution type. Several open connected lines were revolved around the vertical axis with 180° to generate the quarter of the cell wall in a form of shell. The geometry of the nanospikes was also similar to a capsule with one hemispherical end (see Figure 4d). Several closed connected lines were revolved fully around the vertical axis. The cell quarter and the nanospikes were produced separately, and then assembled together (see Figure 5a). Three arrays of nanospikes each with six spikes were developed. The developed nanospikes were assumed to be rigid and solid. 

### 3.2. Setup the Simulation

A static—general (standard implicit) procedure type was selected as the step 1. To make the study quasi-static, the simulation was run for one second while the cell membrane interact with the nanospike under turgor pressure with a non-linear deformation condition. In the incrimination section of this module, 100,000 was given as the maximum number of increments, with a size of 0.01, 0.1, and 0.00001 for the initial, maximum, and minimum increments, respectively. The interaction module was enabled, and a surface-to-surface contact was created, which defining the nanospikes as the master (i.e. rigid and fixed location) and the outer surface of cell wall as the slave (i.e. neither rigid, nor fixed e location) surface. The finite sliding surface-to-surface discretization method was applied. The hard contact for the pressure-overclosure and a default constraint enforcement method were selected for the normal behaviour, and the penalty friction formulation with a friction coefficient of 0.2 was defined for the tangential behaviour. 

### 3.3. Assigning the Boundary Conditions

Refer to Figure 5a. From the property module of the ABAQUS software package, the isentropic elastic type was chosen for the cell wall. The turgor pressure, which is the hydrostatic pressure inside the cell that pushes the membrane of plasma against the wall of the cell, was given a value of 0.03 MPa with an assumed uniform distribution along the cell wall. In addition, an elastic behaviour was assumed for the cell wall when it is modeled under the applied turgor pressure and in contact with the nanospikes. It was assumed that the cell wall would be constrained along horizontal plane (*y*- and *z*-axes) when in contact with the nanospikes, and can only translate along the vertical plane (*x*-axis), which can only be in contact with the nanospikes when the turgor pressure is applied. The same type of boundary conditions were defined for the hemispheric edge of the cell wall. The far-spike membrane edge of the cell wall was restricted in all six degrees of freedom and, therefore, locked against any translation or rotation along or around XYZ axes, such that:*U*_1_ = *U*_2_ = *U*_3_ = *UR*_1_ = *UR*_2_ = *UR*_3_ = 0;(3)
where, *U* and *R* are translational and rotational unit vectors, respectively.

The cylindrical cell membrane edge was defined as the symmetry along the *y*-axis (YSYMM), and the remaining edges along the *z*-axis (ZSYMM). The nanospikes were assumed to be fixed and rigid for all translations and rotations when interacting with the cell wall. 

### 3.4. Grid Generations and Grid Independence Test

Refer to Figure 5b. For the cell wall, a global seed with a sizing control of 0.01 was assigned to form a node for every 0.01 μm along the cell edges. The standard element library was selected from the element type panel with a linear geometric order. As the cell wall was produced as a shell and was expected to experience a large strain deformation with respect to its total size and dimensions, the S4R mesh type was selected, as this has a quad shape, stress/displacement shell element with reduced integration, and a large-strain formulation. Since the nanospikes were produced to be discrete, rigid, and solid, their geometry was first converted into the shell element, and then a global size of 0.01 was given for their seeding to produce mesh with of the R3D4 type. The cell wall consisted of around 18,500 S4R elements, which was around 500 elements for each nanospike. The size was gradually reduced from 0.01 to 0.001 μm, generating five different mesh sizes from which the maximum von Mises stress could be calculated (see Table 2; the trend of the calculated stress is shown in Figure 6). The calculated maximum von Mises stress was found to have increased by only 0.7% for grid system 5. Therefore, for accuracy, the global element size was maintained at 0.001 μm for the rest of the simulation. 

### 3.5. Validation of the Simulation Model with Literature

Using the selected grid system, the maximum von Mises stress and strain were calculated and compared with published results in [61], as shown in Figure 7. Maximum absolute errors in the stress (see Figure 7a) and strain (see Figure 7b) were calculated to be about 6% and 10%, respectively, which is quite acceptable for a finite element model. Figure 7c shows the uniform distribution of the turgor pressure (0.03 MPa) inside the cell wall for which the cell quarter was modelled. Therefore, the simulation can be accepted as validated.

## 4. Conclusions

In order to investigate the rupture mechanism of a bacterial cell on a nanotextured surface, an *E. coli* bacterial cell on a 3 × 6 array of nanospikes was simulated using the three dimensional finite element technique of the ABAQUS/CAE2017 software package. A quarter of an *E. coli* cell was modeled under a specific turgor pressure, and deformation of the cell wall when it had adhered to the nanospikes was observed and analyzed. Since the cell behaves like an elastic material, a linear behavior was assumed between the stress and strain parameters spatially, and being a biological element, those parameters including the strain energy and friction dissipation were assumed nonlinear temporally for the cell membrane at the contact point.

By comparing the obtained results with the published literature, a close agreement with a simulation error of less than 10% was obtained. Based on the simulation results, the following conclusions can be made: The bacterial cell wall is deformed on the location of nanospike tips as full contact is generated under the applied turgor pressure.The location of maximum stress and strain values on the cell wall was observed on the three-phase contact point on which a common region between the cell wall, nanospike, and liquid of the bacterial cell exists.The generated stress on the cell wall due to the applied load is less than the cell tensile strength, it continues to cause a creep deformation, and gradually a rupture can be expected in the cell at around the contact point. The rupture is similar to that of a paper punching mechanism.

A nanotextured surface with nanospikes has the capacity to rupture the cell membrane of an attached bacteria at the contact point, leading to the death of the cell; however this is only theorized in the open literature is not yet proven experimentally. To understand the actual killing mechanism, an extensive study is required, which might be possible computationally. The model and the preliminary results of this study pave the way to an extensive study to explore the mechanism, and potentially to an efficient and productive nanostructured surface for antibacterial application.

## Figures and Tables

**Figure 2 molecules-28-02184-f002:**
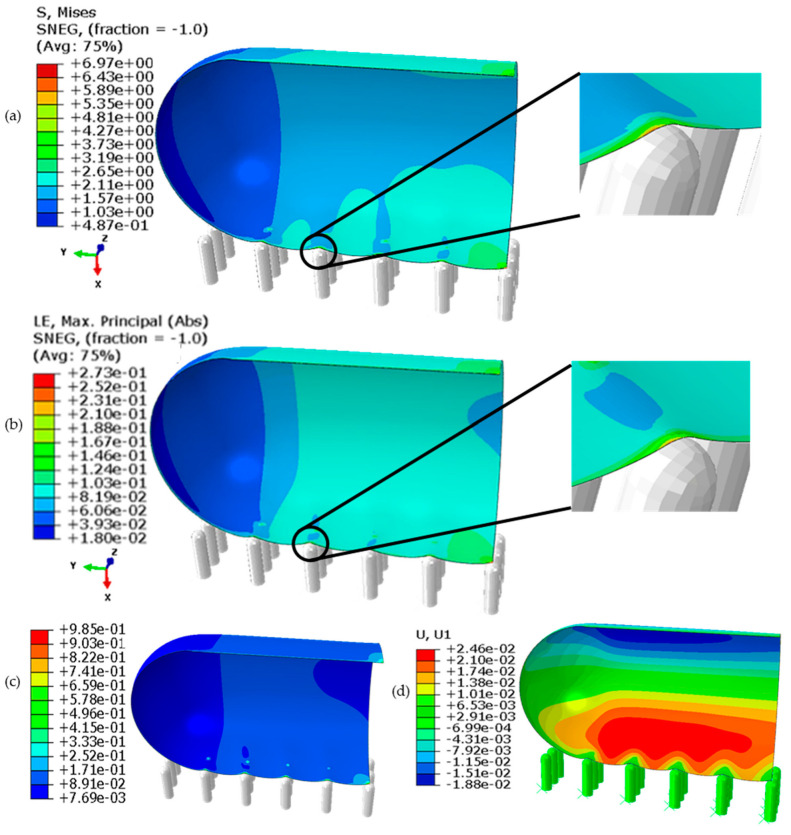
Contour plots of the cell quarter on the nanospikes extracted from the model: (**a**) the von Mises stress (MPa), (**b**) the von Mises strain, (**c**) elastic strain energy density (MPa) and (**d**) displacement along the *x*-axis.

**Figure 3 molecules-28-02184-f003:**
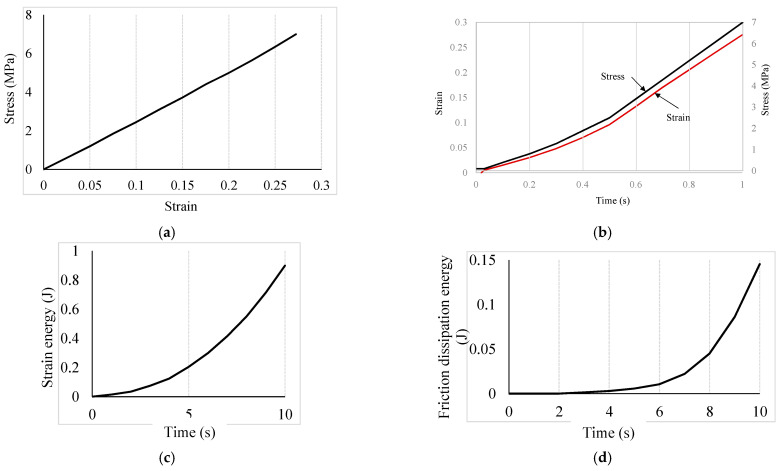
Stress, strain, and temporal dissipated energy trends: (**a**) spatial stress and strain variations; (**b**) temporal stress and strain variations; (**c**) strain energy over time; and (**d**) friction dissipation energy at the contact point.

**Figure 4 molecules-28-02184-f004:**
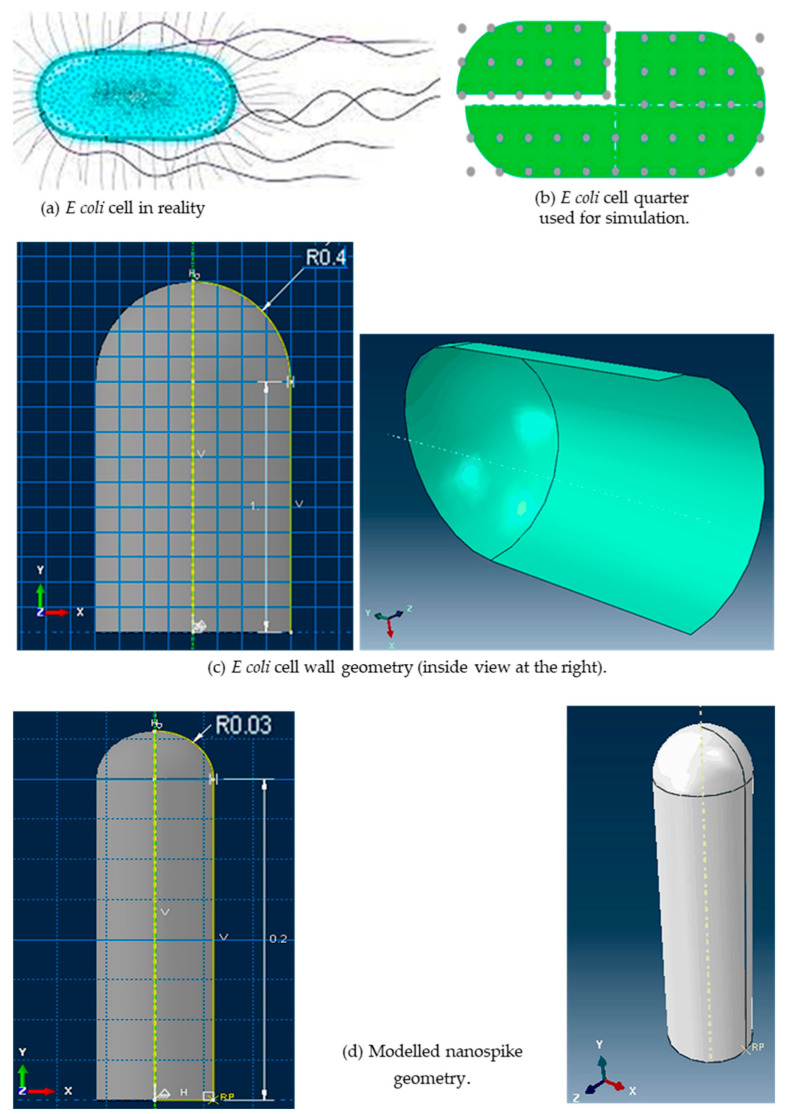
*E. coli* cell in reality [61], and modelled *E. coli* cell and the nanospike.

**Figure 5 molecules-28-02184-f005:**
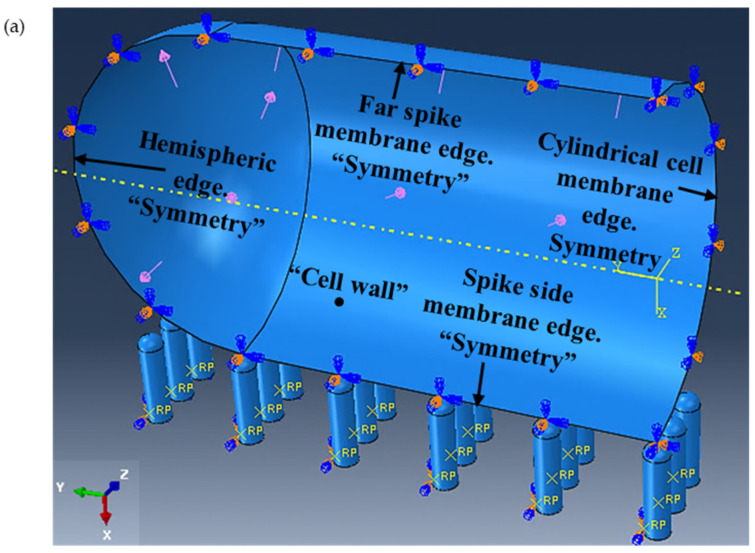
(**a**) Assembly of cell quarter (inside view) and the nanospikes with applied load and boundary conditions, and (**b**) generated mesh.

**Figure 6 molecules-28-02184-f006:**
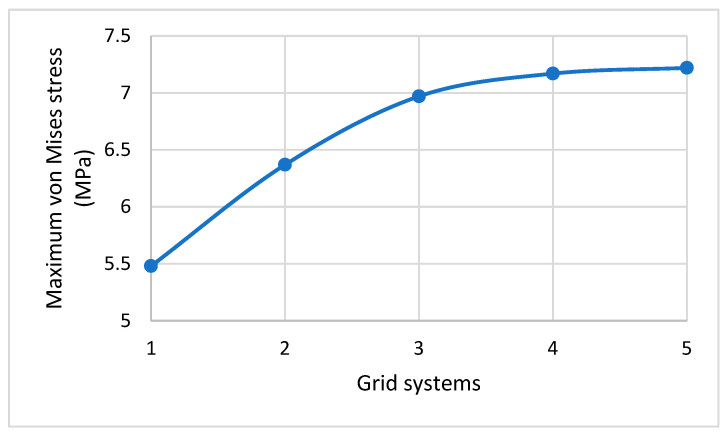
Effect of increasing the total mesh elements on the maximum von Mises stress.

**Figure 7 molecules-28-02184-f007:**
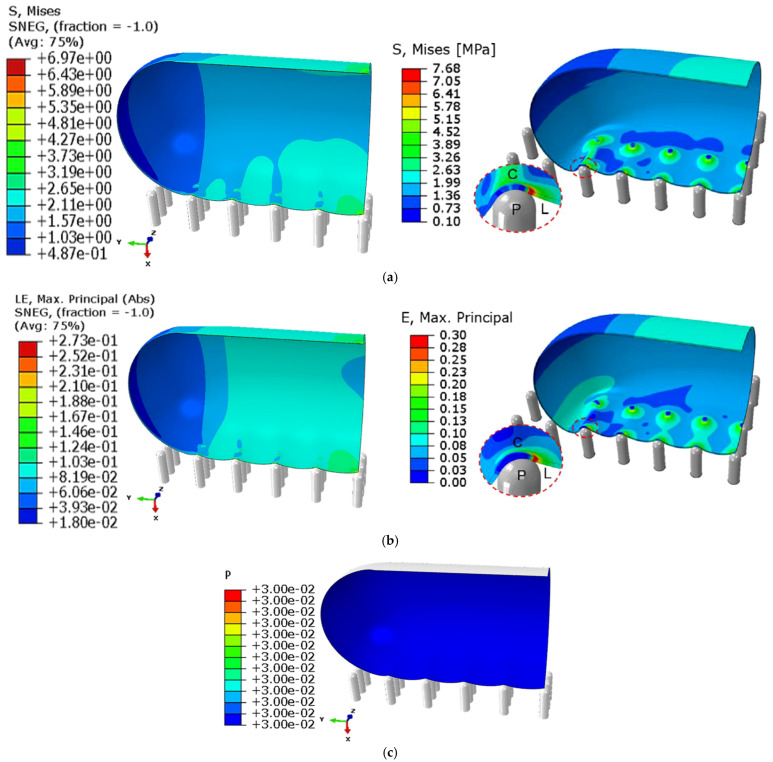
Validation of the obtained simulation results (left) with published results in the open literature by Cui et al. (2021) [61] (right). (**a**) The von Mises stress (MPa) calculated in this model (left) and in the reference (right). (**b**) The von Mises strain calculated in this model (left) and in the reference (right). (**c**) Pressure distribution (in MPa) inside the cell quarter.

**Table 1 molecules-28-02184-t001:** Parameters used for modeling the cell and the nanospike [61].

Modelled cell details	
Cell	Gram-negative, *Escherichia coli* (*E. coli*)
Cylindrical length	1 μm
Radius of the hemispherical ends	0.4 μm
Wall thickness	0.006 μm
Modelled cell mechanics	
Turgor pressure	0.03 MPa
Adhesion strength	6.5 kPa
Critical elastic strength	5 MPa
Tensile strength	13 MPa
Young modulus	25 MPa
Poisson’s ratio	0.16
Modelled nano-spikes details	
Cylindrical length	0.2 μm
Radius of hemispherical end	0.03 μm
Center to center spacing	0.2 μm
Nanospike array	3 × 6

**Table 2 molecules-28-02184-t002:** Details for the mesh independency study.

Grid Systems	Global Size (μm)	No. of Grid Elements	Simulation Time (min.)	Max. von Mises Stress MPa	Stress Increase (%)
1	0.01	18,500	11	5.48	-
2	0.007	37,800	20	6.37	16.24
3	0.005	73,800	90	6.97	9.42
4	0.003	107,600	240	7.17	2.87
5	0.001 Selected	143,500	380	7.22	0.7

## Data Availability

All generated data for this study are included in the article. However, supplementary data, particularly, the computational model belongs to the first and the second authors of this article for reference and future investigations.

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
