# Peer review of "Finite Element Modelling of a Gram-Negative Bacterial Cell and Nanospike Array for Cell Rupture Mechanism Study"

_molecules, 2023, doi:10.3390/molecules28052184_

Round 1

Reviewer 1 Report

This manuscript entitled "Finite element modeling of a gram-negative bacterial cell and nanospike for cell rupture mechanism study" developed a finite element model to study the interaction mechanism between the cell membrane of a bacteria and nanospike at a contact point. The authors compared their model with published data, obtaining a good agreement between the model and published results. This manuscript was well-thought but not well-executed rigorously, unfortunately, it has to be rejected due to the following reasons:

  1. The interaction analysis results and discussion were far from sufficient.
  2. The quality of all the figures has to be improved, including but not limited to (i) the pixel of some figures was too low, (ii) some shape outlines were covered, (iii) font size and style were not consistent, and (iv) explaination of figure contents should be included in the figure legend.
  3. References were not linked.
  4. There is no Table 5 as mentioned in page 9.

Author Response

Please see the cover letter as attached. How the reviewers' comments were addressed is explained at the end of the letter to the editor.

Author Response

Please see the attached cover letter to the editor. At the end of the letter, I discussed how the comments were addressed.

Round 2

Reviewer 1 Report

The authors responded all the comments of the reviewer, therefore, it is okay to publish in Molecules.